# THE DATA MANIFOLD UNDER THE MICROSCOPE

## ABSTRACT

A significant gap exists between theory and practice in deep learning. One example is given by generalization and approximation error bounds, which are often derived for overly simplified models or yield guarantees that are too loose to be informative. Many such bounds rely on the manifold hypothesis and depend on geometric regularity properties, including intrinsic dimension, curvature, and reach of the data manifold or target functions. To make progress on improving these bounds, one needs detailed insight into data manifold geometry and suitable benchmarks on simple datasets. However, existing datasets and analysis tools typically fall into two extremes: analytically defined manifolds with precisely known geometry but limited realism, or real-world datasets where bounds are assessed only through downstream performance and geometric properties can be estimated only coarsely and with hard-to-quantify error.

To address this lack of simple yet realistic datasets and accompanying geometric tools, we introduce a benchmarking framework for studying data geometry. We repurpose and extend the dSprites and COIL-20 datasets with additional transformation dimensions and finer sampling resolution. This enables accurate finite-difference estimates of geometric quantities such as curvature, reach, and volume, yielding a flexible benchmark for evaluating manifold learning methods. As illustrative applications, we assess two established manifold learning bounds by Genovese et al. and Fefferman et al., and analyze how manifold geometry evolves across network layers in $\beta$-VAEs. Our results highlight both the limitations of existing bounds and the value of controlled benchmarks for guiding future theoretical developments.

## 1 INTRODUCTION

Deep learning has become the dominant paradigm for a broad range of tasks, and in recent years generative models in particular such as variational autoencoders (VAEs) (Kingma & Welling, 2013), diffusion models (Ho et al., 2020), and modern masked or autoencoding architectures (He et al., 2022) have seen striking empirical success. A common way to interpret this success is through the manifold hypothesis (Cayton et al., 2005): high-dimensional data often concentrate near low-dimensional manifolds, and learning amounts to finding useful parameterizations of them. Many modern models can thus be viewed as procedures for fitting or approximating data manifolds, whether explicitly, as in latent-variable models (Arvanitidis et al., 2017), or implicitly, as in denoising and score-based flows (Horvat & Pfister, 2021).

While appealing, it remains unclear how networks fit manifolds in practice. Classical results such as Genovese et al. (2012) on minimax rates for manifold estimation, Fefferman et al. (2016) on testing the manifold hypothesis, or more recent work on approximation of Sobolev classes on manifolds (Tan et al., 2024) highlight the role of geometric quantities like curvature, reach, or sampling density in learning. Yet in realistic data these quantities are rarely observable: the data-generating process is unknown, the intrinsic dimension is uncertain (Campadelli et al., 2015), and sampling can be irregular (Shroff et al., 2011; Sedghi et al., 2020). As a result, theoretical guarantees often rely on constants that cannot be directly checked or measured.

This creates two complementary needs. On the theoretical side, sharper and more data-adaptive bounds are desirable, or at least a clearer picture of when existing bounds are informative. On the empirical side, there is a lack of benchmarks that balance realism with geometric control: synthetic manifolds are too simple, while real-world datasets obscure the geometry entirely.

To narrow this gap, we introduce a framework that constructs low-dimensional image families sampled densely along controlled transformation axes (e.g., rotation, translation, scale), and provides efficient finite-difference estimators for geometric measures such as curvature, reach, and volume. Coupled with an experimental pipeline for probing manifold fitting methods (e.g., $\beta$-VAEs), our framework enables systematic tests of how theory and practice align. While in this paper we illustrate its use through generative models and manifold-fitting bounds, the framework is general and can support investigations in other settings, such as discriminative learning or benchmarking geometric measure approximation methods. Our contributions are the following:

- A reproducible framework of adapted low-dimensional datasets with dense, axis-aligned sampling, plus an experimental pipeline for probing modern generative and representation models.
- A suite of efficient finite-difference estimators for pointwise and global geometric quantities such as the curvature, reach, or the volume of manifolds.
- Empirical studies that demonstrate how the framework can be used to test theoretical bounds and trace how learned representations reshape data geometry.

## 2 RELATED WORK

**Manifold Analysis:** Following the manifold hypothesis (Cayton et al., 2005), many works analyze the geometry of data and learned representations. Early results on inferring topological structure from samples include guarantees for homology recovery (Niyogi et al., 2008). Studies on VAEs and $\beta$-VAEs (Kingma et al., 2019; Higgins et al., 2017) show that learned latent spaces often exhibit limited curvature (Arvanitidis et al., 2017; Shao et al., 2018). Geometric invariants—curvature, tangent spaces, and reach—have been used to study robustness and disentanglement (Aamari et al., 2019; Berenfeld et al., 2022; Birdal et al., 2021), though outcomes depend strongly on dataset, architecture, and estimator (Brahma et al., 2015; Kaufman & Azencot, 2023). Recent representation-learning approaches explicitly impose manifold structure via learned charts (Schonsheck et al., 2019). Connections to expressive power have also been explored through manifold topology (Yao et al., 2024a).

**Manifold Fitting Bounds:** Genovese et al. (2012) established the minimax rate $O(n^{-2/(2+d)})$, later shown optimal by Kim & Zhou (2015), though the corresponding estimator is computationally infeasible. Yao & Xia (2019) handled unbounded noise via projection, yielding $O(\sigma^2 \log(1/\sigma))$ Hausdorff error for sample size $O(\sigma^{-(d+3)})$. Neural estimators (Yao et al., 2023; 2024b) offer scalable alternatives at the cost of weaker guarantees. Earlier complexity results for testing the manifold hypothesis (Narayanan & Mitter, 2010) relate geometry, dimension, and sample efficiency more explicitly.

A complementary line of work uses reach as a structural primitive. Fefferman et al. (2016) introduced a geometric approach based on preserving reach, with noisy-data extensions reducing complexity from double- to single-exponential (Fefferman et al., 2018; 2020). However, these guarantees depend on unknown geometric parameters such as reach and volume. For example, the constants in Yao & Xia (2019) scale as $\tau^{-2}$, illustrating a recurring issue: theoretical bounds often require prior knowledge of quantities that can only be estimated from the manifold itself.

**Geometric Property Estimation:** Recent work develops estimators and sample complexity bounds for reach and related quantities (Aamari et al., 2019; Berenfeld et al., 2022; Aamari et al., 2023), though empirical validation remains challenging due to the lack of ground-truth geometric data. Similar progress exists for curvature estimation: scalar curvature estimators with nonasymptotic guarantees (Aamari & Levrard, 2019; Gawlik & Neunteufel, 2025), methods for the second fundamental form and Ricci curvature (Acosta et al., 2023; Samal et al., 2018), and tangent-space estimators with provable accuracy (Cheng & Chiu, 2016; Cazals & Pouget, 2005). These results extend the classical geometric recovery literature (Niyogi et al., 2008).

Overall, theory provides strong asymptotic guarantees, but empirical understanding is hindered by the absence of datasets with exact geometric ground truth. Our framework addresses this gap by combining dense, axis-aligned sampling with finite-difference estimators, enabling controlled evaluation of geometric quantities and systematic tests of how theoretical bounds behave under realistic approximations.

## 3 BACKGROUND AND DEFINITIONS

**Datasets as Manifolds:** The manifold hypothesis suggests that high-dimensional datasets often concentrate near low-dimensional manifolds. We focus on datasets where the number of intrinsic factors of variation $d$ is small and fixed, and where these factors are explicitly known. Each dataset is modeled as a union of smooth $d$-dimensional manifolds embedded in $\mathbb{R}^D$, possibly with boundary. Typically, each of the $k$ semantic classes in the dataset corresponds to a separate connected component of this union.

For this work, we restrict attention to simple topologies in which each manifold factors into cyclic and non-cyclic directions. Concretely, every manifold is assumed to be homeomorphic to $[0,1]^r \times (S^1)^s, r + s = d$, so that some coordinates vary over a compact interval while others wrap around a circle. This setting captures many common synthetic datasets. A canonical example is *dSprites*, a collection of $64 \times 64$ grayscale images of objects (square, ellipse, heart) undergoing controlled transformations such as scaling, rotation, and translation. The manifold of a class in this dataset is homeomorphic to $[0,1]^3 \times S^1$ and is embedded into $\mathbb{R}^{4096}$.

To obtain discrete datasets from this continuous geometric picture, we impose a grid structure. Let $n_l$ be the number of sampled values along the $l$-th dimension, so that the total number of grid points is $n = \prod_{l \leq d} n_l$. We define

$$G = \left\{ \left( \tfrac{j_1}{n_1}, \ldots, \tfrac{j_r}{n_r}, 2\pi \tfrac{j_{r+1}}{n_{r+1}}, \ldots, 2\pi \tfrac{j_d}{n_d} \right) \,\middle|\, 0 \leq j_\ell < n_\ell \right\}, \tag{3.1}$$

where the first $r$ coordinates sample $[0,1]$ uniformly and the last $s$ coordinates sample $S^1$ uniformly. For each class $i \leq k$, we define a mapping, $u_i : G \to M_i$, that associates each grid point with a dataset element obtained by applying the corresponding transformations. The image $u_i[G]$ provides a discrete parametrization of the manifold $M_i$, and by cutting along cyclic directions one obtains discrete patches of $M_i$. The complete discretized dataset is then

$$X_G = \bigcup_{i \leq k} u_i[G]. \tag{3.2}$$

**Geometric Measures:** We focus on three geometric quantities describing the local and global structure of a manifold: *volume*, *scalar curvature*, and *reach*. For completeness, we recall the definitions of the Riemannian metric and the objects needed to introduce these quantities.

**Definition 3.1** (Riemannian metric)**.** A Riemannian metric $g$ on a $d$-dimensional differentiable manifold $M$ assigns to each point $p \in M$ an inner product $g_p$ on the tangent space $T_pM$. If $M$ is embedded in $\mathbb{R}^D$, the ambient Euclidean metric induces a Riemannian metric by restriction. In local coordinates $u : U \to M$, the metric matrix is

$$g_{ij}(x) = \langle \partial_i u(x), \partial_j u(x) \rangle_{\mathbb{R}^D}. \tag{3.3}$$

**Definition 3.2** (Volume element and volume)**.** On a Riemannian manifold $(M, g)$ with local coordinates $(u^1, \ldots, u^d)$, the natural volume element is

$$d\mathrm{V} = \sqrt{\det(g)} \, du^1 \wedge \cdots \wedge du^d. \tag{3.4}$$

The volume of a region $R \subset M$ is then

$$\mathrm{Vol}(R) = \int_{u^{-1}(R)} \sqrt{\det(g)} \, dx^1 \cdots dx^d. \tag{3.5}$$

Throughout, we use Einstein summation notation: free upper and lower indices in the same term are implicitly summed, e.g. $g^{ij} Ric_{ij} = \sum_{i,j} g^{ij} Ric_{ij}$.

**Definition 3.3** (Notions of curvature)**.** The curvature of $(M, g)$ is encoded in the Riemann tensor

$$R^i{}_{jkl} = \partial_k \Gamma^i_{jl} - \partial_l \Gamma^i_{jk} + \Gamma^i_{kr} \Gamma^r_{jl} - \Gamma^i_{lr} \Gamma^r_{jk}, \tag{3.6}$$

where the Christoffel symbols are

$$\Gamma^i_{jk} = \tfrac{1}{2} g^{ir} (\partial_j g_{rk} + \partial_k g_{rj} - \partial_r g_{jk}). \tag{3.7}$$

Contracting yields the Ricci tensor $Ric_{ij} = R^r{}_{irj}$ and the scalar curvature $R = g^{ij} Ric_{ij}$.

**Definition 3.4** (Reach). For a closed set $A \subset \mathbb{R}^D$, the *medial axis* is the set of points with more than one nearest neighbor in $A$. The *reach* of $A$ is the distance to its medial axis:

$$\tau_A = \inf_{p \in A} d_{\ell^2}(p, Med(A)). \tag{3.8}$$

Here $d_{\ell^2}$ is the Euclidean distance: $d_{\ell^2}(p, q) = \|p - q\|_2$, and for a set $A$, $d_{\ell^2}(p, A) = \inf_{q \in A} d_{\ell^2}(p, q)$. Intuitively, $\tau_A$ is the largest radius such that every point within distance $< \tau_A$ of $A$ has a unique nearest neighbor in $A$. For a compact $d$-dimensional submanifold $M \subset \mathbb{R}^D$, the reach can be expressed as

$$\tau_M = \inf_{p \neq q \in M} \frac{\|p - q\|^2}{2\, d_{\ell^2}(q - p, T_p M)}, \tag{3.9}$$

where $d_{\ell^2}(v, T_p M)$ is the distance of a vector $v$ to the tangent space at $p$.

In our applications, we consider both *global* quantities (total volume, the scalar curvature integral[1], global reach) and their *local* counterparts (volume element, pointwise scalar curvature, and local reach estimators as in Aamari et al. (2019)).

**Definition 3.5** (Hausdorff distance). Let $A, B \subset \mathbb{R}^D$ be non-empty subsets. The *Hausdorff distance* between $A$ and $B$ is

$$H(A, B) = \max \left\{ \sup_{a \in A} d_{\ell^2}(a, B), \ \sup_{b \in B} d_{\ell^2}(b, A) \right\}. \tag{3.10}$$

If $M$ is the ground-truth data manifold and $\hat{M}$ a learned approximation (e.g., via an autoencoder), then $H(M, \hat{M})$ expresses the largest geometric error, i.e. how far the manifolds can be from each other at any single point.

**Manifold Fitting and Bounds:** Manifold learning is closely connected with generative modeling: in both cases, high-dimensional data are assumed to lie near a low-dimensional manifold. Generative models typically learn a mapping between the data manifold in $\mathbb{R}^D$ and a lower-dimensional latent space where semantic factors of variation are disentangled and sampling is easier. If the encoder–decoder network is sufficiently regular (locally $\mathcal{C}^1$ or $\mathcal{C}^2$), then composing the encoder with the manifold charts $u_i$ yields a parametrization of the latent manifold. Each data point is mapped to its latent representation, and decoding corresponds to projecting back onto the estimated data manifold.

We focus on two theoretical results that provide bounds on the error of manifold fitting in terms of sample size, intrinsic dimension, and geometric properties of the manifold. Before stating these results, we define the normal fiber of radius $r$ at a point $p \in M$ as $L_r(p) = T_p^\perp M \cap B_D(p, r)$, where $T_p^\perp M$ is the space orthogonal to the tangent space at $p$ and $B_D(p, r)$ is the $D$-dimensional ball of radius $r$ centered at $p$. Intuitively, this fiber is a $(D - d)$-dimensional ball extending away from the manifold at $p$.

**Minimax manifold estimation bound:** Genovese et al. (2012) establish minimax rates for estimating a manifold from samples corrupted by small normal noise. Their result shows that the intrinsic dimension $d$ alone governs the difficulty of estimation.

**Theorem 1** (Genovese et al. (2012)). Let $\mathcal{M}(\tau)$ be the class of compact, smooth, boundaryless $d$-dimensional manifolds embedded in $\mathbb{R}^D$ with reach at least $\tau$. For each $M \in \mathcal{M}(\tau)$, let $Q_M$ be the distribution of $Y = \xi + Z$, where $\xi$ is uniformly distributed on $M$ and $Z$ is uniformly distributed on the normal fiber of radius $\sigma$ at $\xi$. Let $\mathcal{Q} = \{Q_M \mid M \in \mathcal{M}(\tau)\}$. For an $n$-sample estimator $\widehat{M} : \mathbb{R}^n \to \mathcal{M}$, define the minimax risk

$$R_n(\mathcal{Q}) = \inf_{\widehat{M}} \sup_{Q \in \mathcal{Q}} \mathbb{E}_Q[H(\widehat{M}, M)], \tag{3.11}$$

where $H(\cdot, \cdot)$ denotes the Hausdorff distance. Then there exist constants $C_1, C_2 > 0$ such that

$$C_1 \left( \frac{1}{n} \right)^{\frac{2}{2+d}} \leq R_n(\mathcal{Q}) \leq C_2 \left( \frac{\log n}{n} \right)^{\frac{2}{2+d}}. \tag{3.12}$$

---

[1]The integral of the scalar field defined by the scalar curvature $R$ on $M$ over the volume of $M$, i.e. $\int_M R\, dV$.

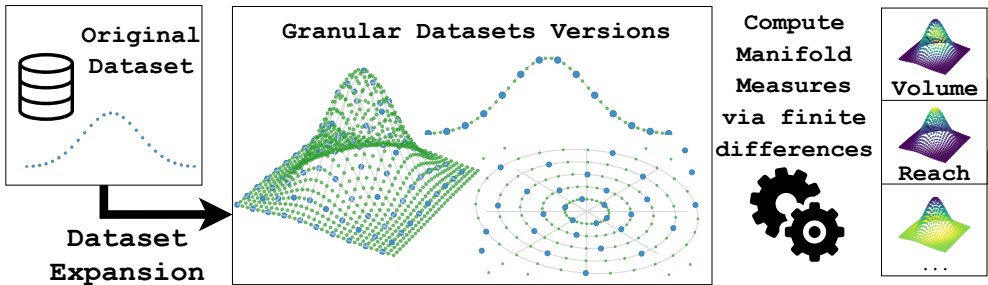

Figure 1: From a low-dimensional seed we produce dense, regular grids via analytic parametrizations or systematic transforms; central finite differences recover geometric estimators such as reach and curvature enabling validation of manifold-fitting bounds.

This result shows that the sample complexity depends exponentially on the intrinsic dimension $d$, but not on the ambient dimension $D$. Intuitively, the distributions $Q_M$ correspond to sampling a point on $M$ and perturbing it orthogonally within its reach. In our setting, an estimator $\widehat{M}$ may be interpreted as a neural network (e.g., an autoencoder) that outputs a fitted manifold.

**Testing the manifold hypothesis:** Fefferman et al. (2016) introduce a testing framework for the manifold hypothesis. Their analysis relates the required number of samples to two geometric parameters of the manifold: its volume and its reach. Assuming a manifold $M$ satisfying the hypothesis exists, they derive the following upper bound on the Hausdorff distance between the true and estimated manifolds:

$$R_n(\mathcal{Q}) \leq C_1 \frac{V^{1/d}}{\tau} \left(\frac{1}{n}\right)^{1/d}. \tag{3.13}$$

Further details on the derivation of this bound are provided in Appendix A.

## 4 METHODS AND FRAMEWORK

We construct controlled synthetic manifolds of low intrinsic dimension ($d = 1$–$4$) sampled on regularly spaced grids. Dense sampling enables stable finite-difference approximations of partial derivatives, allowing accurate computation of the induced metric, volume element, curvature tensors, and reach. This framework provides a setting to empirically assess theoretical manifold-fitting bounds and a reproducible dataset suite for validating geometric estimators.

**Datasets with a dense grid structure:** We consider two complementary approaches for constructing low-dimensional datasets with a grid structure.

For manifolds with an explicit mathematical description, we use known parametrizations to generate a regular grid of points. For image-based or domain-specific datasets, we extend existing data by systematically applying transformations (translations, rotations, scalings). Sampling all combinations of these transformations yields a structured grid analogous to the analytic manifold case.

Although grid sampling provides dense coverage, it may not be uniform with respect to the intrinsic geometry. To obtain more uniform subsets, we use two strategies:

- Iterative farthest-point sampling on the grid, selecting maximally separated points.
- Sampling according to the volume form, either by weighting grid points proportionally or, when analytic expressions exist, by using inverse transform sampling across dimensions via numerical integration. This becomes computationally demanding as $d$ increases.

For experiments, we construct a fixed test set by selecting a uniformly distributed subset of grid points. The remaining points form the training set, with varying training sizes. This ensures clear train/test separation while keeping the test set sufficiently representative for approximating Hausdorff and average distances between manifolds. We compute the geometric measures on the full dataset.

**Computation of geometric measures:** To estimate geometric quantities such as the volume element, scalar curvature, and reach, we use finite-difference approximations of partial derivatives on the dense grid. We assume $\mathcal{C}^3$ regularity for estimating the reach and volume and $\mathcal{C}^5$ regularity for estimating scalar curvature, enabling second-order finite-difference approximations of the required first- and third-order derivatives.

Given a parametrization $u : \mathbb{R}^d \to \mathbb{R}^D$, the central difference approximation

$$f'(x) = \frac{f(x+h) - f(x-h)}{2h} + O(h^2) \tag{4.1}$$

extends coordinate-wise. For example,

$$D_{h,i}^c u(x) = \frac{u(x_1, \ldots, x_i + h, \ldots, x_d) - u(x_1, \ldots, x_i - h, \ldots, x_d)}{2h}, \tag{4.2}$$

gives a second-order accurate estimate of the $i$th partial derivative. Products of such approximations yield $g_{ij} = \langle u_{,i}, u_{,j} \rangle + O(h^2)$, so all smooth metric-dependent quantities (volume form, Christoffel symbols, curvature tensors, scalar curvature) inherit $O(h^2)$ accuracy. These intermediate tensors are available within the framework for downstream analysis. Thus $\hat{V} - V = O(h^2)$ and $\hat{R} - R = O(h^2)$. For the reach, we use the estimator of Aamari et al. (2019), which on an equidistant grid of spacing $h$ satisfies

$$|\hat{\tau} - \tau| = O(h). \tag{4.3}$$

We validate these estimators on manifolds with known closed-form geometric quantities, including 2- and 3-dimensional ellipsoids, hyperboloids, and 4-spheres in various ambient spaces. Across all tested cases, relative errors are typically below $10^{-2}$, confirming that finite differences on dense grids provide highly accurate geometric approximations.

**Comparison to general estimation methods:**

General estimators of curvature and reach from point clouds, such as those of Aamari et al. (2023) for reach and Aamari & Levrard (2019) for curvature, operate in far broader settings than ours. They must infer derivatives from unstructured samples via sophisticated interpolations, leading to optimal but more complex rates, nontrivial constants, and substantial computational overhead. No publicly available implementations currently exist.

On a $d$-dimensional grid with $n$ points, spacing is $h = n^{-1/d}$. Our reach estimate therefore satisfies

$$|\hat{\tau} - \tau| = O(h) = O(n^{-1/d}),$$

which matches the lower bound and is slightly sharper than the $O((\log n/n)^{1/d})$ rate of Aamari et al. (2023) for $\mathcal{C}^3$ manifolds.

For scalar curvature, our intrinsic finite-difference estimator gives

$$|\hat{R} - R| = O(h^2) = O(n^{-2/d}).$$

The bound of Aamari & Levrard (2019) is $O((\log n/n)^{3/d})$ for $\mathcal{C}^5$ manifolds, which is slightly better. This gap is due to our intrinsic computation of the Riemann curvature tensor, which requires third-order derivatives. An extrinsic formulation via the second fundamental form would reduce the derivative order and yield optimal rates. This is a natural extension for future work.

Our aim is not to compete with general estimators. Instead, the controlled grid setting provides a straightforward and reproducible way to compute geometric quantities at near-optimal accuracy. The resulting measures serve as reliable benchmarks and unit tests for developing and analyzing more general methods on unstructured data.

## 5 Applications

**Datasets:** Our experimental setup employs two types of datasets: (i) toy manifolds with explicit mathematical parametrizations for analytical validation, and (ii) adapted versions of established image datasets (dSprites Matthey et al. (2017) and COIL-20 Nene et al. (1996)) that provide controlled settings for empirical evaluation. Table 1 summarizes the key characteristics of each dataset.

The toy manifolds serve as analytically tractable benchmarks with known closed-form geometric properties. For the image datasets, we modified the original sampling schemes to better suit geometric computations: dSprites was regenerated with reduced resolution but improved anti-aliasing, while COIL-20 was extended with additional transformations including scale and orientation. Both datasets employ dense grid sampling with margin oversampling on non-cyclic dimensions to enable finite difference computations of geometric measures. Detailed specifications are in Appendix B.

**Manifold fitting methods:** We consider two complementary approaches to fit manifolds to the datasets: a geometric method, *Manifold Moving Least Squares* (MMLS) (Sober & Levin, 2020) , and a deep learning method, the $\beta$-VAE autoencoder (Higgins et al., 2017).

*MMLS:* MMLS, Sober & Levin (2020), is a local manifold approximation method in which, for each query point, a neighborhood is weighted by a kernel function and a $d$-dimensional affine space together with a local polynomial are fitted by weighted least squares. The projection onto the approximated manifold is then obtained by projecting onto the fitted $d$-plane and evaluating the polynomial correction, yielding a flexible higher-order local model of the manifold.

In our setting, the ambient dimension is too high to reliably estimate local polynomials. We therefore customize the procedure by restricting the fit to the weighted $d$-plane alone. There are more details on the method and our usage in Appendix C.1.

*$\beta$-VAE:* The $\beta$-VAE, Higgins et al. (2017), provides a learning-based approach. It maps input data into a low-dimensional latent space and reconstructs them, with the reconstruction lying on a learned manifold. The reconstruction distance can thus be interpreted as the distance to this manifold. As the input dimensionality is too small for meaningful bottlenecks in toy datasets, we employ $\beta$-VAE only for image datasets (dSprites, COIL-20). For more details, refer to Appendix C.1

**Sampling protocol:** For both methods, a fixed test set of 500 uniformly spread points is used. On the toy datasets, training sets range from 5–500 points, and each experiment is repeated 20 times. On dSprites and COIL-20, data ratios range from 0.01 up to 1.0, with three repetitions per setup.

The motivation for considering both methods is that MMLS directly fits a manifold in the original data space, whereas $\beta$-VAE learns a latent manifold with richer semantic structure. Together, they provide complementary views of manifold fitting performance.

**Error Bounds evaluation:** We now turn to the first application of manifold fitting: the empirical evaluation of theoretical bounds on approximation error. Recall from section 3 that Genovese et al. provide an upper and lower bound depending only on the intrinsic dimension, while Fefferman et al. derive a sharper upper bound that additionally incorporates the reach and volume of the manifold. For the fitting methods (MMLS, $\beta$-VAE), each dataset (toy, dSprites, COIL-20), and a range of intrinsic dimensions, we fit manifolds for multiple training set sizes. As distance metric we use the Hausdorff distance between the fitted and the true manifold.

**Comparing to bounds:** To estimate the constants appearing in the theoretical upper and lower error bounds, we avoid attempting to compute these constants directly. Instead, we determine them empirically by aligning the theoretical shapes of the bounds with the observed error curve.

The theoretical bounds scale like $C_1 n^{-2/(2+d)}$ and $C_2(\log n/n)^{-2/(2+d)}$, which suggests that the true error curve should be well approximated by a power law of the form $R_n \sim C n^{-g(d)}$. Taking logarithms yields a linear model $\log R_n \sim -A \log n + \log C$. We therefore regress $(\log n_j, \log \hat{R}_{n_j})$ for a sequence of sample sizes $n_j$ covering the fractions $\{0.01, 0.02, 0.05, 0.1, \ldots, 1.0\}$ of the test

Table 1: Dataset characteristics and parametrizations used in our experiments:

| Dataset | $d$ | $D$ | Factors | Range | Components |
|---|---|---|---|---|---|
| Circle | 1 | 2 | $\theta$ | $[0, 2\pi]$ | 1 |
| Two moons | 1 | 2 | $\theta$ | $[0, 2\pi]$ | 2 |
| Sphere | 2 | 3 | $\theta, \phi$ | $[0, 2\pi] \times [0, \pi]$ | 1 |
| Torus | 2 | 3 | $\theta, \phi$ | $[0, 2\pi] \times [0, 2\pi]$ | 1 |
| dSprites | 4 | 4096 | scale, orientation, pos. x, pos. y | see Appendix B | 3 |
| COIL-20 | 3 | 4096 | horiz. orient., scale, img. orient. | see Appendix B | 20 |

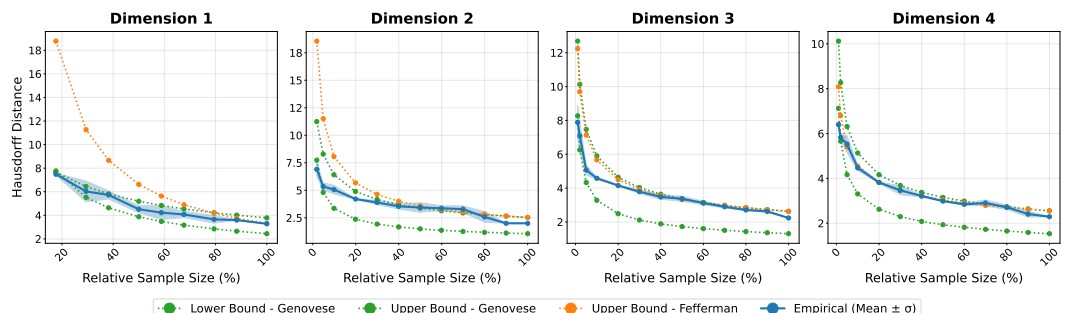

Figure 2: Fitting bounds on dSprites for different dimensions utilizing MMLS.

dataset. Each experiment is repeated three times, and the empirical values $\hat{R}_{n_j}$ used for the fit are the pointwise averages. The regression provides estimates $\hat{A}$ and $\hat{C}$, giving a fitted curve $\hat{R}_n^{\text{fit}} = \hat{C}n^{-\hat{A}}$.

We then use this fitted curve to select constants for the upper and lower theoretical bounds. For each $n_j$, we compute the 0.99 percentile of the fitted values $\hat{R}_{n_j}^{\text{fit}}$. The upper bound constant is chosen as the smallest value for which the bound stays above these percentiles; the lower bound constant is chosen as the largest value for which the bound stays below them. This procedure yields stable and reproducible constants while preserving the theoretical scaling behavior. For more details and plots related to the logarithmic regression fitting, please look at the appendix sub-section C.2.

**Results:** We present three representative comparisons:

- **Cross-dataset** (Figure 4): curves for sphere, torus, COIL-20 and dSprites using MMLS.

- **Cross-dimension** (Figure 2): curves for dSprites with intrinsic dimensions $d = 1, 2, 3, 4$, again using MMLS.

- **Cross-model** (Figure 3): comparison of MMLS and $\beta$-VAE on 4D dSprites.

The main observations are: (i) Fefferman's bounds, which explicitly exploit reach and volume, yield significantly tighter upper envelopes than dimension-only bounds. (ii) For toy datasets, the empirical curves initially follow the upper rate $\left(\frac{\log n}{n}\right)^{1/d}$ but then approach the lower rate $\left(\frac{1}{n}\right)^{1/d}$ as sample size grows. (iii) For image datasets (COIL-20, dSprites), the behavior is

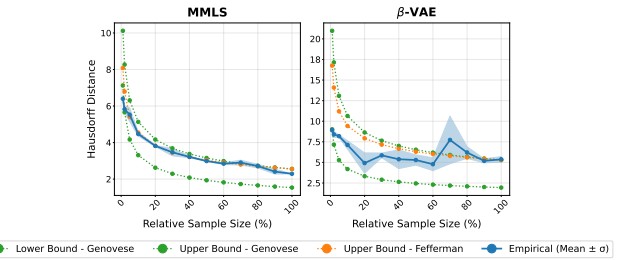

Figure 3: Bounds for MMLS & $\beta$-VAE on dSprites (4D)

reversed: curves start closer to the lower bound and move upward with larger $n$. We actually notice that this behavior is correlated with the increase of scalar curvature magnitude on the datasets which captures their complexity. One potential line of investigation could be to check if the $\log n$ factor on the upper bounds can be connected with curvature or reach.

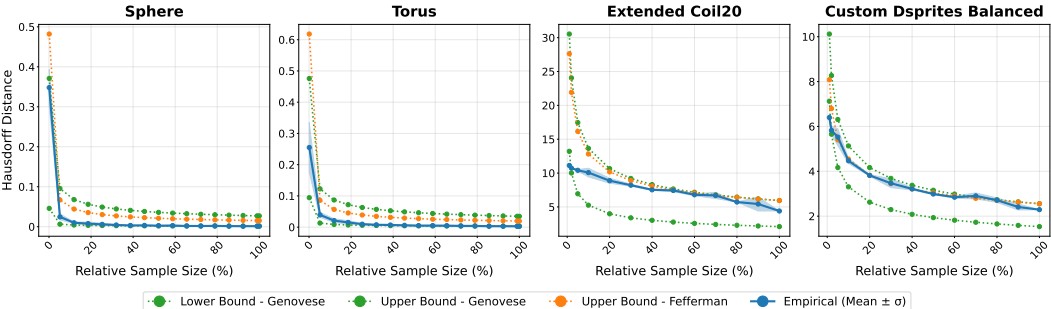

Figure 4: Fitting bounds for MMLS on from left to right Sphere, Torus, COIL-20 and dSprites.

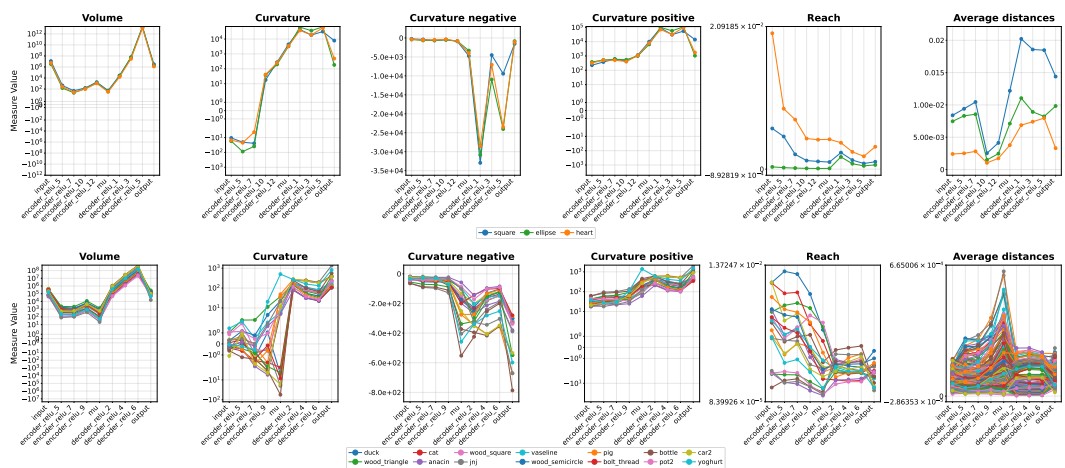

Figure 5: Evolution of volume, scalar curvature, and reach of the manifold across layers of a $\beta$-VAE, shown for both dSprites and COIL-20:

**Manifold analysis:** In the second application, we analyze how geometric properties of data manifolds evolve through the layers of a $\beta$-VAE using the dSprites (4D) and COIL-20 (3D) datasets. For each layer we compute the manifold volume, integrated scalar curvature (with positive and negative parts separated), reach, and the average distance between class manifolds (Figure 5).

We observe that curvature systematically increases while reach decreases in deeper layers, indicating that the intermediate manifolds become progressively more intricate and approach self-intersections. At the same time, average distances between class manifolds grow closer to the latent space, suggesting that as semantic information strengthens, the network prioritizes class separation over preserving low-level visual transformations.

# 6 DISCUSSION

We presented a framework for constructing dense manifolds and accurately estimating their geometric properties, enabling controlled studies of manifold fitting. We illustrated two use cases: assessing existing manifold fitting bounds via log-log scaling fits, and analyzing how data geometry evolves across layers of a $\beta$-VAE. Both revealed how geometric structure influences learning.

The main strength of this framework is its ability to provide ground truth geometric quantities that are otherwise inaccessible or unreliable. Unlike real datasets, where assumptions cannot be verified, or simple analytic manifolds, which lack representational richness, our synthetic constructions balance realism and control. This makes them well-suited for probing theoretical assumptions, validating estimators, or stress-testing bounds under controlled perturbations.

Several limitations should nonetheless be emphasized:

- The framework can analyze only manifolds of low dimension, up to 4–5. It is therefore intended for benchmarking and understanding rather than analyzing arbitrary datasets.
- It currently supports manifolds with simple topology, $[0, 1]^r \times (S^1)^s$. Extending this is possible but comes with additional implementation complexity.

Future work could address these limitations in several directions. Expanding the dataset suite to include richer transformations, occlusions, more diverse topologies, and additional modalities (e.g., text or audio) would broaden applicability. Another possibility is to evaluate existing geometric estimators, e.g. for curvature or reach, by comparing them against the more accurate finite-difference values on the test manifolds. The framework could also be used to study discriminative bounds, where classifiers effectively fit functions on manifolds, and to quantify how geometric fidelity interacts with generalization.

Finally, large language models (LLMs) were used to polish the writing of this paper. All scientific ideas, results, and analyses are original.

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

## A    BOUNDS FROM TESTING THE MANIFOLD HYPOTHESIS FEFFERMAN ET AL. (2016)

Fefferman et al. (2016) provide a testing framework for the manifold hypothesis. Their result connects the number of samples to geometric parameters of the manifold, namely its volume and reach. Here we present a fuller version of their result which implies the bound mentioned in the main text.

**Theorem 2** ( Fefferman et al. (2016))**.** There exists an algorithm which, given samples from a distribution $P$ and $\varepsilon > 0$, distinguishes with probability at least $1 - \delta$ between the following two cases:

- There exists $M \in G(d, CV, \tau/C)$ such that $\mathcal{L}(M, P) \leq C\varepsilon$.

- There does not exist $M \in G(d, V/C, \tau C)$ such that $\mathcal{L}(M, P) \leq \varepsilon/C$.

Here $G(d, V, \tau)$ denotes the class of $d$-dimensional manifolds of volume $\leq V$ and reach $\geq \tau$. The required sample size is

$$n \;=\; \frac{N_p \, \ln^4(N_p/\varepsilon) + \ln(\delta^{-1})}{\varepsilon^2}, \qquad N_p = V\left(\frac{1}{\tau^d} + \frac{1}{\tau^{d/2}\varepsilon^{d/2}}\right). \tag{A.1}$$

Assuming that such a manifold $M$ exists, the result yields an upper bound on the Hausdorff distance between the true and estimated manifolds:

$$R_n(\mathcal{Q}) \;\leq\; \frac{C_1}{\tau}\left(\frac{V}{n}\right)^{1/d}. \tag{A.2}$$

## B    DATASETS

### B.1    DSPRITES

For the original dataset check Matthey et al. (2017). Some example images can be seen in figure 6 and a projection of dataset to 3D with PCA in 7. One can notice how the projections of the manifolds of the different classes are enveloped.

**Parameters:**

- Image shape: $64 \times 64$
- Shape: 3 values (square, ellipse, heart)
- Scale: 16 values linearly spaced in $[0.5, 1]$
- Orientation: 16 values in $[0, 2\pi]$
- Position X: 16 values in $[0, 1]$
- Position Y: 16 values in $[0, 1]$
- Total: 196,608 images ($3 \times 16^4$)

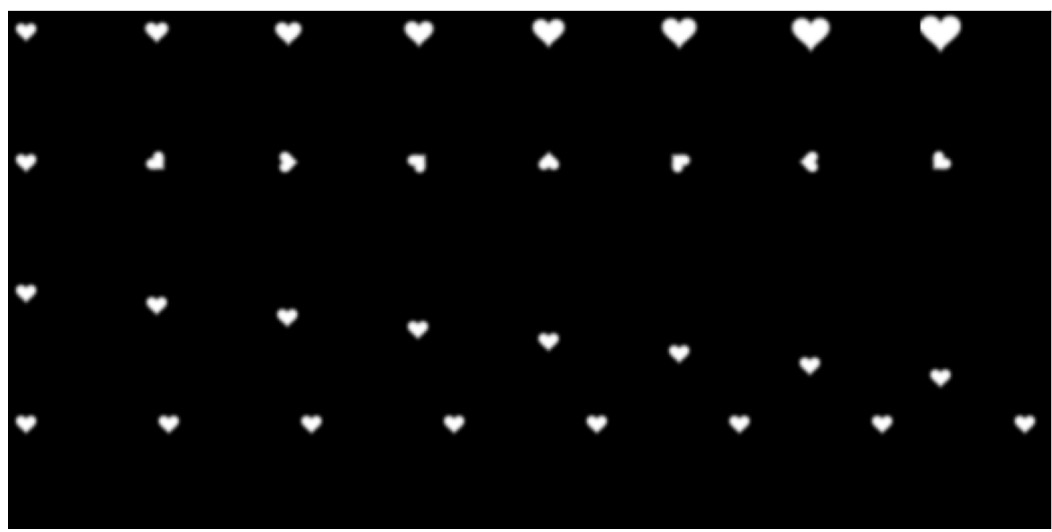

Figure 6: Example images of the heart class of dSprites.

## B.2 COIL-20

For the original dataset check Nene et al. (1996). Some example images can be seen in figure 8 and a projection of dataset to 3D with PCA in 9. Again one can notice how the classes are enveloped.

**Parameters:**

- Image shape: $64 \times 64$
- Objects: 20 objects from original COIL-20
- Horizontal orientation: 18 values in $[0, 2\pi]$
- Scale: 16 values linearly spaced in $[0.5, 1]$
- Image orientation: 16 values in $[0, 2\pi]$
- Total: 92,160 images ($20 \times 18 \times 16^2$)

## B.3 GEOMETRIC MEASURES ON PROJECTIONS

Below we also provide two 3D plots where the geometric measures are plotted on the hear class of dSprites 10 and on the duck class of COIL-20 11. The absolute scalar curvature, is simply the absolute value of the point-wise scalar curvature. One can notice in both datasets that the absolute scalar curvature aligns well with the reach while the usual scalar curvature is much more noisy. This indicates that the different transforms have an intense impact on the change of curvature direction of each other.

## Classes

- square
- ellipse
- heart

Figure 7: PCA projection of the dSprites dataset.

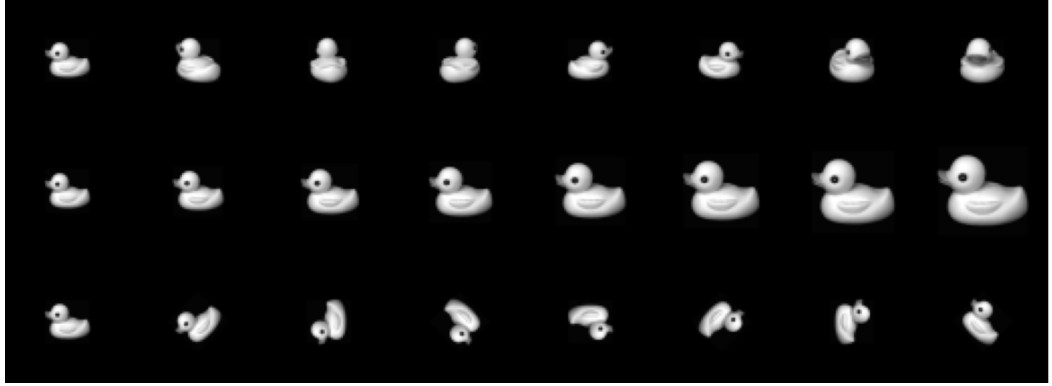

Figure 8: Example images of the duck class of COIL-20. Note the resizing and the additional yz-rotation and size transforms.

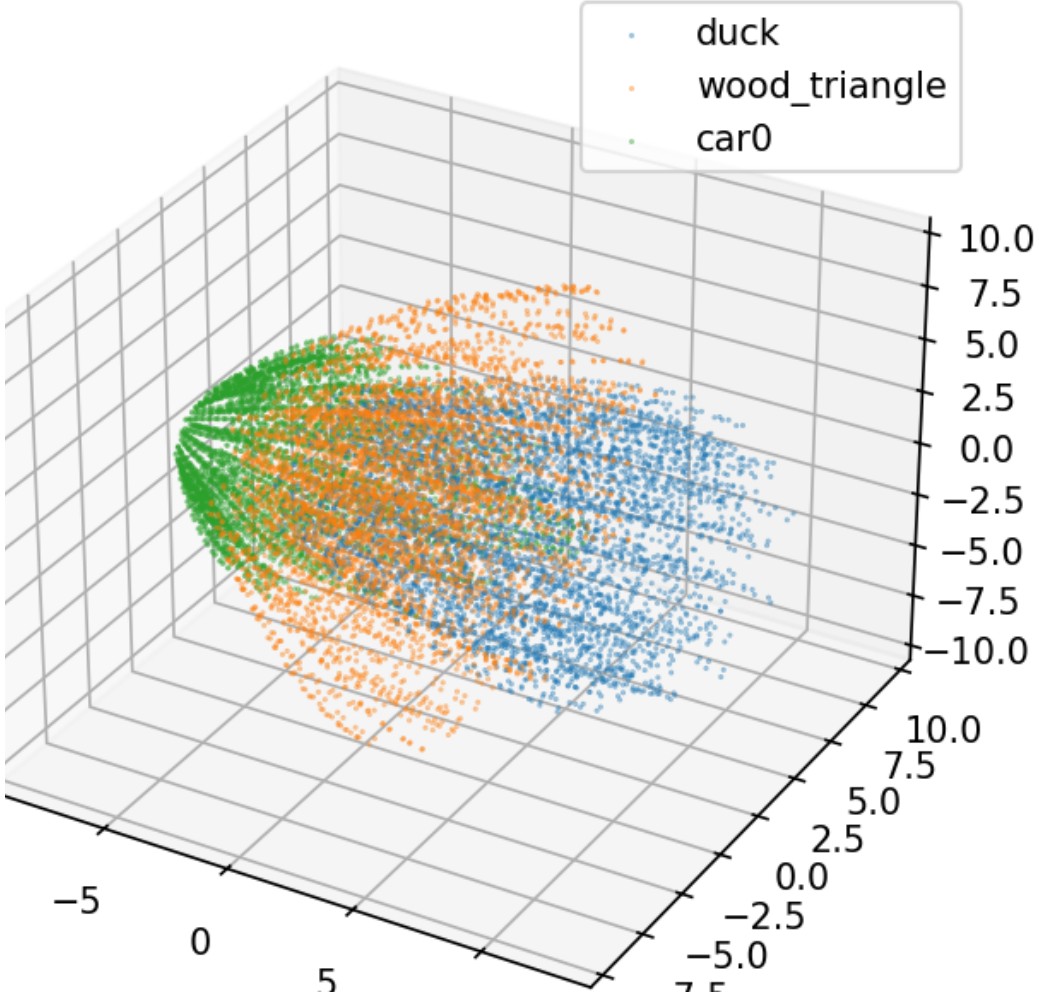

Figure 9: PCA projection of the COIL-20 dataset.

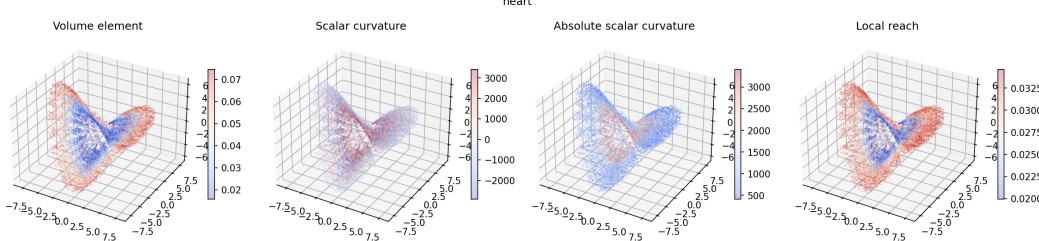

Figure 10: PCA projection of the COIL-20 dataset.

## C  MANIFOLD FITTING

Here we provide details on the way we setup and run the experiments for the manifold fitting use case of our framework. This included details on the datasets used, the fitting models and the way we estimate the bound curves.

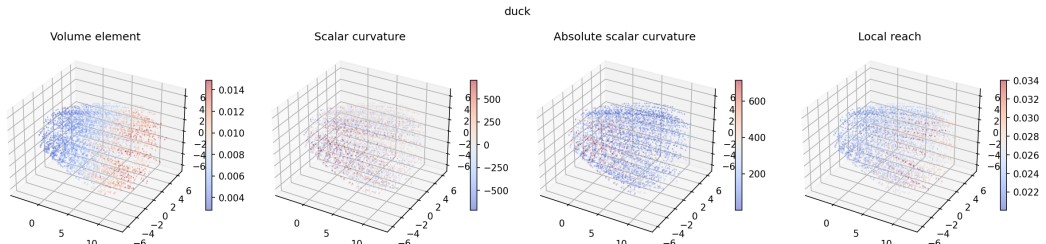

Figure 11: PCA projection of the COIL-20 dataset.

## C.1 FITTING METHODS

### THE FITTING PROCESS AND DATA SELECTION

Most manifold-fitting methods estimate local structures on $M$—such as tangent $d$-planes, tubular neighborhoods, or charts—from which a reconstructed manifold $\widehat{M}$ is obtained. Evaluating the fit ideally requires computing the Hausdorff distance $H(M, \widehat{M})$. A direct numerical approximation demands dense, uniformly distributed point sets on both $M$ and $\widehat{M}$, together with exhaustive nearest-neighbor searches, which is computationally prohibitive.

For simple analytic manifolds, uniform coverage can be obtained through explicit parametrizations, either via grids or by uniform volume sampling. For more complex datasets, curated sampling is often needed to empirically cover the space. To avoid evaluating the full Hausdorff distance, one can use the projection operators provided by the fitting methods: for each point $p \in M$, let $\widehat{p}$ be its projection onto $\widehat{M}$. The projection error $\|p - \widehat{p}\|_2$ serves as a surrogate for $d_{\ell^2}(p, \widehat{M})$, and $H(M, \widehat{M})$ is approximated by the maximum error over a dense reference set on $M$.

**Toy manifolds.** Our procedure is:

- Construct a dense, approximately uniform reference set $Y = \{y_i\}_{i \leq n_{\text{test}}} \subset M$.
- Uniformly sample $n$ points $X = \{x_i\}_{i \leq n} \subset M$ for fitting.
- Fit the chosen manifold-fitting method on $X$.
- Project $Y$ to $\widehat{Y}$ and compute the pointwise errors $\|y_i - \widehat{y}_i\|_2$ and their maximum.

The dense reference set $Y$ is generated either directly via a grid in a closed-form parametrization or by drawing a large uniform sample $S = \{s_i\}_{i \leq N} \subset M$ and selecting $n_{\text{test}}$ centroids $C = \{c_i\} \subset M$ that minimize the Sinkhorn loss between $C$ and $S$. All uniform sampling steps rely on closed-form sampling formulas tailored to each manifold.

**Image manifolds.** The procedure mirrors the synthetic case:

- Select a dense subset $Y = \{y_i\}_{i \leq n_{\text{test}}} \subset X_G$ from the grid $X_G$.
- Uniformly sample $X = \{x_i\}_{i \leq n} \subset X_G \setminus Y$ for fitting.
- Fit the manifold-fitting method on $X$.
- Project $Y$ to $\widehat{Y}$ and compute $\|y_i - \widehat{y}_i\|_2$ and their maximum.

Dense subset selection is performed by a maximal-distance sub-sampling algorithm. Uniform sampling on the grid uses a discrete distribution obtained by normalizing the per-point volume element. Projection is geometric for MMLS and via reconstruction for the $\beta$-VAE.

### MANIFOLD MOVING LEAST SQUARES (MMLS)

MMLS Sober & Levin (2020) is a projection-based manifold fitting method. Given sample points $Y = \{y_i\}_{i \leq N} \subset \mathbb{R}^D$ drawn from an unknown manifold $M$, the algorithm estimates a fitted manifold $\widehat{M}$ and provides a projection operator that maps any nearby point $p$ to $\widehat{M}$. A key ingredient is

a kernel $\theta$ that assigns similarity weights based on point–point distances, enabling smooth local approximations.

The procedure consists of two stages:

- **Local affine fitting.** For a query point $p \in \mathbb{R}^D$ close to $M$, find an affine subspace $H$ and a point $q \in B_D(p, \frac{\tau}{2})$ that minimize

$$\sum_{i \leq N} d_{\ell^2}(y_i, H)^2 \, \theta(\|y_i - q\|_2),$$

subject to $p - q \perp H$, where $\tau$ denotes the reach of $M$. This step provides a locally estimated tangent space.

- **Local polynomial reconstruction.** With $q$ and $H$ fixed, fit a multivariate polynomial $g : \mathbb{R}^d \to \mathbb{R}^D$ by solving

$$\sum_{i \leq N} \|g(\mathrm{proj}_H(y_i)) - y_i\|_2^2 \, \theta(\|y_i - q\|_2),$$

where $\mathrm{proj}_H(y_i)$ is the orthogonal projection onto $H$. This serves as a local analogue of the exponential map, taking coordinates in the tangent space $T_q\widehat{M}$ and mapping them to $\widehat{M}$. The final projection of $p$ is then $g(p)$.

The method originates from the Moving Least Squares framework for hypersurfaces Levin (2003) and was later extended to arbitrary embedded manifolds Sober & Levin (2020). Other techniques exist (e.g., Zhang & Zha (2004) and the constructions used in Fefferman et al. (2016; 2018); Genovese et al. (2012)), but these are typically more intricate or tailored to dimensionality reduction rather than direct manifold fitting.

**Implementation for our experiments.** We apply MMLS using the dense grid subset $Y$ as the reference set and the uniformly sampled points $X$ as queries. Each $x \in X$ is projected onto $\widehat{M}$ as follows:

- Identify the $k$ nearest neighbors $\mathcal{N}_x = \{y_{i_1}, \ldots, y_{i_k}\} \subseteq Y$.
- Compute distances $\|x - y_{i_m}\|_2$ for all neighbors and assign weights via an isotropic Gaussian kernel with bandwidth $\sigma$:

$$w_m = \exp\left(-\left(\frac{\|x - y_{i_m}\|_2}{2\sigma}\right)^2\right).$$

- Estimate $q$ as the weighted average of $\mathcal{N}_x$. Estimate the affine space $H$ by performing weighted PCA on $\mathcal{N}_x$ with weights $w_m$.
- Project $x$ orthogonally onto $H$. In place of the full polynomial stage in Sober & Levin (2020), we use a degree-1 (local linear) approximation.

**Hyperparameters.**

- $k = 5$ for toy manifolds and $k = 2^{d+1}$ for image manifolds.
- Gaussian kernel bandwidth: $\sigma = 1.0$.

$\beta$-VAE

We use the reference implementation from WonKwang Lee (2018), which provides two standard architectures: model **B** from Higgins et al. (2017) and model **H** from Burgess et al. (2018). Both operate on $64 \times 64$ images and differ mainly in channel width and bottleneck design.

**Model B.**

Encoder: Conv $32 \times 4 \times 4$ (stride 2), $32 \times 4 \times 4$ (stride 2), $32 \times 4 \times 4$ (stride 2), $32 \times 4 \times 4$ (stride 2), FC $256 \to 256 \to 2z_d$.

Decoder: FC $z_d \to 256 \to 256 \to 512$, followed by Deconv layers mirroring the encoder (stride 2).

All layers use ReLU activations.

**Model H.** Encoder: Conv $32 \times 4 \times 4$ (stride 2), $32 \times 4 \times 4$ (stride 2), $64 \times 4 \times 4$ (stride 2), $64 \times 4 \times 4$ (stride 2), $256 \times 4 \times 4$ (stride 1), FC $2z_d$.

Decoder: FC $z_d \rightarrow 256$, followed by Deconv layers reversing the encoder (strides 1 and 2).

All layers use ReLU activations.

**Objective.** Model **H** uses the standard ELBO with a strengthened KL term:

$$\mathcal{L} = \text{recon} + \beta \, D_{\text{KL}}, \qquad \beta > 1.$$

Model **B** follows the capacity-controlled formulation:

$$\mathcal{L} = \text{recon} + \gamma \, |D_{\text{KL}} - C|,$$

where $C$ is annealed linearly from 0 to $C_{\max}$.

**Hyperparameters.**

- $\beta = 4$, $\gamma = 100$, $C_{\max} = 20$.
- Learning rate: $5 \times 10^{-4}$ (dSprites), $10^{-4}$ (COIL-20).
- Batch size: 64.
- Latent dimension: $z_d = 10$.
- Optimizer: Adam with $\beta_1 = 0.9$, $\beta_2 = 0.999$.
- Architecture choice: model B for dSprites, model H for COIL-20.
- Training budget: up to $10^6$ iterations or $10^5$ epochs.
- Early stopping when training loss does not improve for $0.5\%$ of max epochs.

These settings closely follow WonKwang Lee (2018), with minimal tuning for stable convergence. After training on the dense set $Y$, the decoder is used to reconstruct the test points $X$.

## C.2 Bound Curves

### The difficulty of computing the bounds' constants

The constants appearing in most bounds papers are unfortunately hard to compute but most importantly very large to be practical. The reason is because the authors goal is to derive the bounds in terms of complexity and not to optimize those constants .To demonstrate how this looks like, we track how such a constant appears in one of the bounds. Take the lower bound by Genovese et al. (2012), this is:

$$C_1 \left(\frac{1}{n}\right)^{\frac{2}{2+d}} \leq R_n(\mathcal{Q}). \tag{C.1}$$

We will use some notation from the paper, if the reader wishes to understand the details, the aforementioned paper should be consulted. To derive this bound, the authors use Le Cam's lemma, which bounds the supremum of errors of an estimator $\widehat{M}$ by the discrepancy between any two estimators $M_0, M_1$ (see proof of theorem 1, section 3.2 in Genovese et al. (2012)):

$$\sup_{Q \in \mathcal{Q}} \mathbb{E}_{Q^n}[H(M, \widehat{M})] \geq H(M_0, M_1) \, ||Q_0^n \wedge Q_1^n||.$$

Note that the manifolds are all in $\mathcal{M}(\kappa)$, where $\kappa$ is the reach of the true data manifold (this is usually denoted by $\tau$, but the authors use $\kappa$ in their paper) and a value $0 < \sigma < \kappa$ is selected as a limit below the reach for the support of the considered distributions. Then the distribution space is $\mathcal{Q} = \mathcal{Q}(\kappa, \sigma) = \{Q_M | M \in \mathcal{M}(\kappa)\}$ where $Q_M$ is the density on $M \oplus \sigma$ defined by uniformly sampling a point $p$ on $M$ and then sampling a point orthogonally to $M$ at $p$ up to distance $\sigma$, or to be precise, on the fiber of size $\sigma$ at $p$, $L_\sigma(p) = T_p(M)^\perp \cap B_D(p, \sigma)$.

Now selecting $M_0$, $M_1$ in a way that they differ as much as possible, given the constraint they should belong in $\mathcal{M}(\kappa)$, yields the following inequality:

$$H(M_0, M_1) \, ||Q_0^n \wedge Q_1^n|| = H(M_0, M_1)(1 - \frac{1}{2}\int |q_0 - q_1|)^{2n} \geq \frac{\gamma}{2}(1 - c\gamma^{\frac{d+2}{2}})^{2n}.$$

Then setting $\gamma$ to $n^{-\frac{2}{d+2}}$ leads to the bound, because for large enough $n$:

$$\frac{\gamma}{2}(1 - c\gamma^{\frac{d+2}{2}})^{2n} = \left(\frac{1}{n}\right)^{\frac{2}{d+2}} \frac{1}{2}(1 - \frac{c}{n})^{2n} \sim \left(\frac{1}{n}\right)^{\frac{2}{d+2}} \frac{1}{2}e^{-c}$$

. The constant $c$ now comes from theorem 6 of the paper where the distance between the density functions $q_0, q_1$ of $Q_0, Q_1$ is $\int |q_0 - q_1| = O(\gamma^{\frac{d+2}{2}})$. Tracing this to the proof of the theorem, in section 7.2, one notices that this comes from the following inequality near the end of the proof:

$$V(S_0 - S_1) \leq C\sigma^{D-d-1}\sqrt{4\gamma\kappa - \gamma^2}^d\gamma,$$

where the constant $C$ accounts for the coefficients of the volumes of the $d$- and $D - d - 1$ balls involved in the product set containing $S_1 - S_0$ and the distortion involved in the product, thus:

$$C = c_1(\kappa, \sigma) \, \omega_d \, \omega_{D-d-1}.$$

1080

Using Steiner's formula, one can bound $c_1(\kappa, \sigma)$ by $(1 - \left(\frac{\sigma}{\kappa}\right)^2)^{-\frac{d}{2}}$. By symmetry, one gets the same inequality for $V(S_1 - S_0)$. If one additionally uses the approximation $\sqrt{4\gamma\kappa - \gamma^2} = 2\sqrt{\kappa\gamma} + o(\gamma)$ and assumes $\gamma$ is small enough we get:

$$V(S_0 \circ S_1) \leq 2\, C \, \sigma^{D-d-1}\sqrt{4\gamma\kappa - \gamma^2}^d 2\gamma \leq \left[\left(1 - \left(\frac{\sigma}{\kappa}\right)^2\right)^{-\frac{d}{2}} 2^{d+2} \, \omega_d \, \omega_{D-d-1} \, \kappa^{\frac{d}{2}} \, \sigma^{D-d-1}\right] \gamma^{\frac{d+2}{2}}$$

Finally, lemma 4 connects distributions $q_M$ on $M \oplus \sigma$ with the volume density on the same set, $u_M$ by: 1026

$$q_M \leq (1 + \frac{\sigma}{\kappa})^d \omega_d u_M,$$

integrating which we get:

$$\int |q_0 - q_1| \leq 2\,(1 + \frac{\sigma}{\kappa})^d \, \omega_d \, \frac{V(S_0 \circ S_1)}{V(M \oplus \sigma)}.$$

We can use the tubular bound $V(M \oplus \sigma) \geq V(M) \, \omega_{D-d}\sigma^{D-d} \, (2 - (1 - \frac{\sigma}{\kappa})^{D-d})$ and end up with the following explicit formula for $C_1$:

$$C_1 = \frac{1}{2} \, e^{-c}$$

where

$$c = 2^{d+3} \frac{\kappa^{\frac{d}{2}}}{\sigma^{D-d}} \frac{(1 - \left(\frac{\sigma}{\kappa}\right)^2)^{-\frac{d}{2}} (1 + \frac{\sigma}{\kappa})^d}{2 - (1 - \frac{\sigma}{\kappa})^{D-d}} \frac{\omega_d \, \omega_{D-d-1}}{\omega_{D-d}}$$

**Takeaways**

- There are hidden dependencies of $C_1$ on $\sigma, \kappa, D, d$ and on $n$, if $n$ is small.

- While estimating a formula for $C_1$ we needed to use separate bounds which make $C_1$ potentially much smaller than it could actually be. Additionally, one needs to select a value for $0 < \sigma < \kappa$, which makes the computation of $C_1$ also a bit arbitrary.

- The lower bound we presented has the simplest derivation of the constant. The two upper bounds we consider are more complicated and involve the use of more intermediate inequalities which make the corresponding constants too large.

ESTIMATING THE BOUNDS' CONSTANTS

Based on the obstacles on directly computing the constants of the bounds, we decided to select them instead based on the values of the empirical curve. The idea is to simply place the upper bounds so that they just touch the empirical curve from above and the lower bounds so that they just touch the empirical curve from below. To get more reliable results, we first fit a curve on the empirical curve and use this to place the bounding curve. The selection of the curves family for the empirical curve is based on the observation that the formulas related to bounds depend exponentially on the dimension and multiplicatively on the constant for example looking at the Genovese et al. upper and lower bounds:

$$
C_1 \left( \frac{1}{n} \right)^{\frac{2}{2+d}} \quad \leq \quad R_n(\mathcal{Q}) \quad \leq \quad C_2 \left( \frac{\log n}{n} \right)^{\frac{2}{2+d}}. \tag{C.2}
$$

It is reasonable thus to assume that the real error curve can be well approximated by a formula of the form:

$R_n \sim C(\frac{1}{n})^{g(d)}$,

which can be written in terms of logarithms as:

$\log R_n \sim -g(d) \, \log n + \log C = -A \, \log n + \log C.$

With this assumption, we use the logarithms of the empirical curve values $\{(\log n_j, \log \hat{R}_{n_j})\}_{j \in J}$, where $\{n_j \mid j \in J\} \subseteq [0, N]$ is an increasing sequence of number of samples of the total $N$ dataset points, to fit the above regression. This way we get estimates $\hat{A}$, $\hat{C}$ of $A$ and $C$. Then we get the resulting fitted empirical curve:

$$
\hat{R}_n^{fit} = \hat{C} \left( \frac{1}{n} \right)^{\hat{A}}.
$$

Finally, using the values of $\hat{R}_n^{fit}$, we set the values of constants of the lower/upper bounds to be the largest/smallest constants such that the corresponding bounds are below/above the 0.99 percentile of the values of $\hat{R}_{n_j}^{fit}$ on the experiment values $\{n_j\}_{j \in J}$. In our experiments, those values correspond to the

$$[0.01, 0.02, 0.05, 0.1, \ldots, 1.0]$$

fractions of the dataset of $N$ points. Finally, note that for each percentage the experiment is repeated three time and the empirical curve used her is their point-wise average.

ANALYSIS OF THE FITTED EMPIRICAL CURVES

To provide a better understanding on the way we select the upper and lower bounds, we display a version of figures 4, 3 and 2 including the fitted empirical curve as a blue dotted line in figures 14, 13 and 12.

Furthermore, we include figures 17, 16 and 15 with the regression line on the logarithmic values of the empirical curves, residual plots on the fitted values, QQ plots and also the values of $R^2$. Those are again available for all three ablations present in the main paper. Most fits look reasonably good, with the exception of the $\beta$-VAE plot which is a bit noisy and the low values on the sphere and torus where on very low numbers of points, 5-20, there seems to be a need for a second, less steep function which to model them.

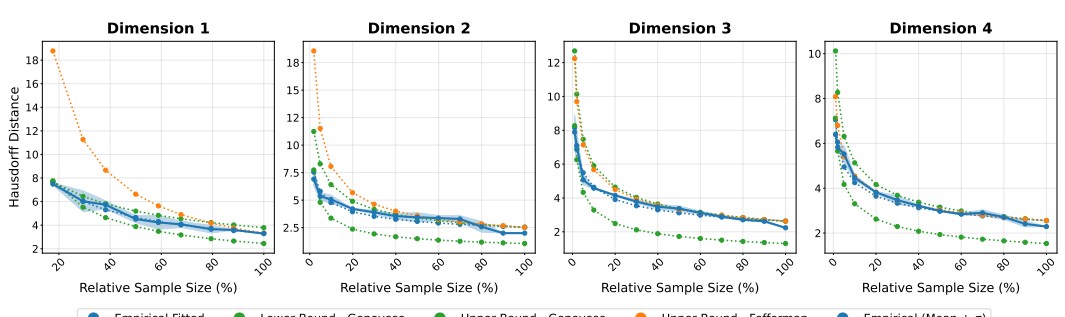

Figure 12: Fitting bounds on dSprites for different dimensions utilizing MMLS.

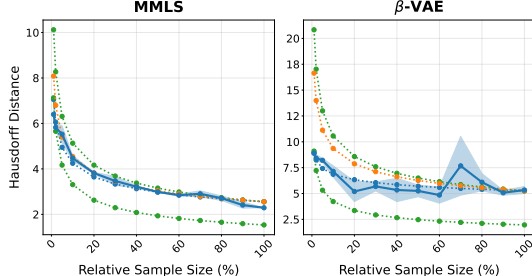

Figure 13: Fitting bounds for MMLS & $\beta$-VAE on dSprites (4D)

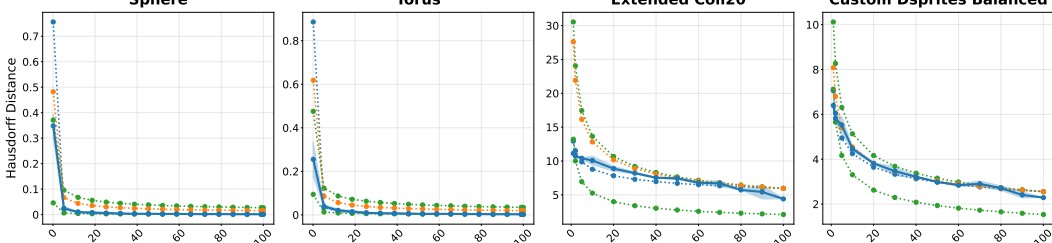

Figure 14: Fitting bounds for MMLS on from left to right Sphere, Torus, COIL-20 and dSprites.

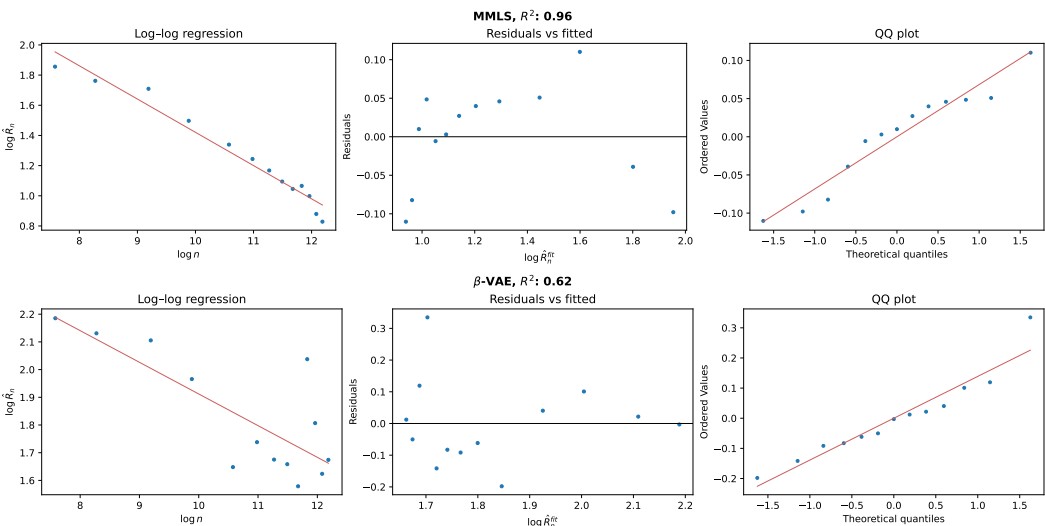

Figure 15: Regression evaluation on dSprites for different dimensions utilizing MMLS.

Figure 16: Regression evaluation for MMLS & $\beta$-VAE on dSprites (4D)

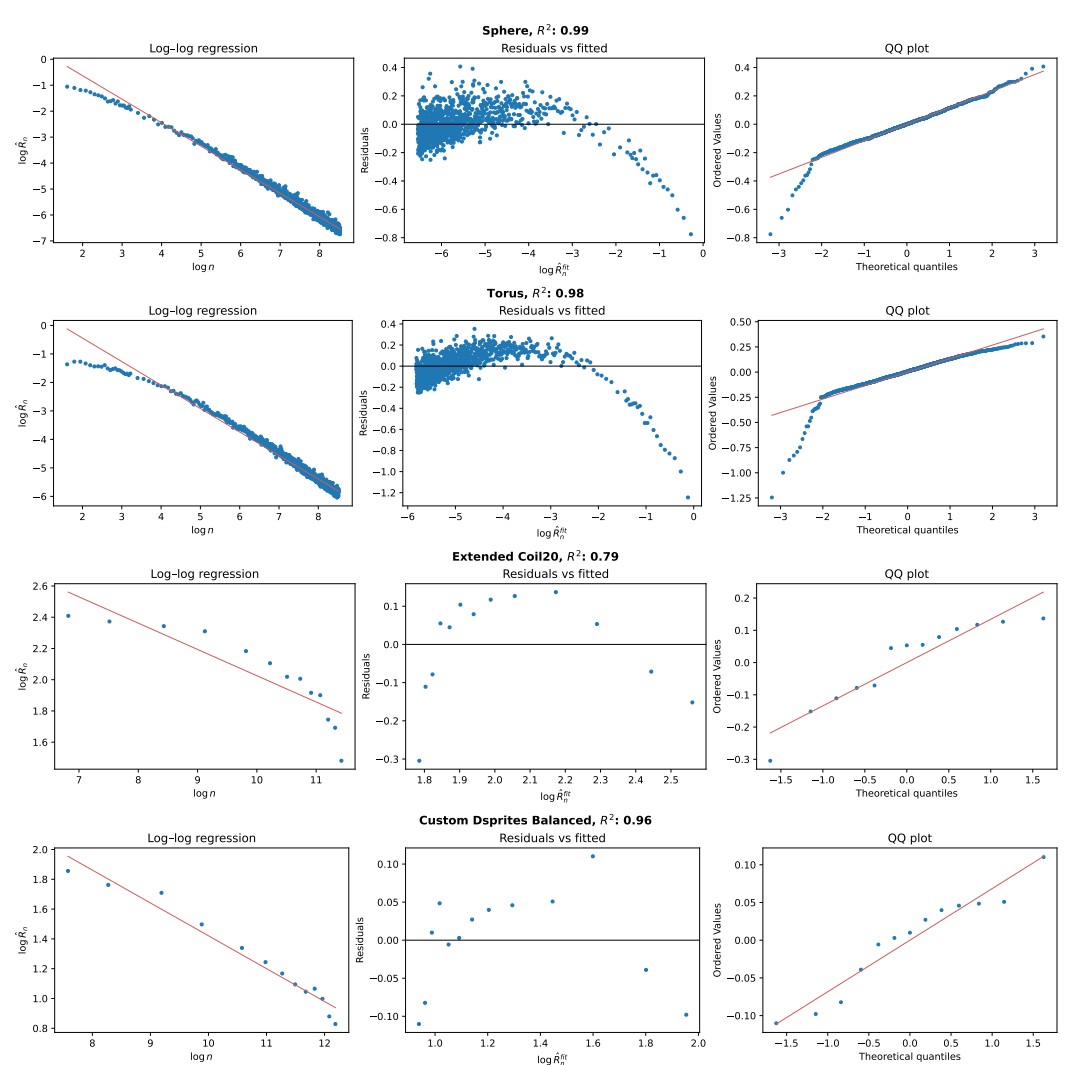

Figure 17: Regression evaluation for MMLS on from left to right Sphere, Torus, COIL-20 and dSprites.

