# OpenReview forum: "The data manifold under the microscope"
_ICLR.cc/2026/Conference — Submitted to ICLR 2026_

### Official Review · Reviewer_tZnq · 2025-10-17

**Soundness:** 3
**Presentation:** 2
**Contribution:** 1
**Rating:** 2
**Confidence:** 4

**Summary:**

This paper purports to bridge the gap between theory and practice in deep learning for data satisfying the manifold hypothesis. The paper is based on empirical studies that aim to test whether theoretical upper (and lower) bounds for manifold approximation are sharp, i.e., overly not overly pessimistic (or optimistic).

The method is based on introducing a uniform sampling grid for simple, low-dimensional manifolds which are homeomorphic to [0,1]^r \times (S^1)^s, r+s=d. They then use existing manifold approximations. They consider a couple data sets and empirically observe results similar to the upper bound in one case and and the lower bound in the other.

Overall, this paper falls short of its goal of bridging theory and practice. There is little new here. Furthermore, the method relies on finite difference schemes which in turn rely on uniform sampling which limits the applicability of this method to real data. The "axis aligned" assumption is also quite limiting. Additionally, for this paper to make a more sizeable contribution, it should either a) introduce new theory or b) provide insights into model performance under complex manifolds, preferably under less than idealized sampling conditions.

Minor

line 43, extra space after hypothesis
equation 3.14 \mathcal{Q} does not appear to be defined

**Strengths:**

The background on differential geometry is mostly well written and there is some decent discussion of relevant literature

**Weaknesses:**

See above

**Questions:**

Do you have any insights as to when the upper or lower bounds will be tighter?

---

> ### Author Response · Authors · 2025-11-26
> **Response to review**
>
> Thank you for your review. We address your concerns below.
>
> **This paper falls short of its goal of bridging theory and practice**
>
> Our primary goal is to provide a practical and reproducible framework that enables empirical evaluation of geometric bounds under the manifold hypothesis. The contribution is not new theory, but rather the infrastructure needed to make such theoretical investigations possible/easier.
>
> We agree that the method is not applicable to large-scale natural datasets and does not aim to overcome the limitations imposed by uniform sampling or axis alignment. Those restrictions are by design: they enable exactness and interpretability, which are central for benchmarking geometric methods. Even with these constraints, the framework provides a valuable resource for researchers who want to:
>
> - Test new geometric estimators.
> - Explore failure modes of manifold-learning techniques.
> - Study how curvature, sampling density, or manifold structure affect algorithmic behavior.
> - Generate new synthetic manifolds with known differential structure.
>
> To our knowledge, there is currently no alternative that offers such controlled conditions together with accurate geometric ground truth.
>
> **line 43**
>
> We have now added the definition of $\mathcal{Q}*
>
>
> **Do you have insights into when the upper or lower bounds will be tighter?**
>
> Our empirical observations suggest the following conjecture:
>
> Upper bounds tend to be tighter for manifolds with larger curvature magnitude or more pronounced geometric complexity.
>
> Lower bounds tend to be tighter for simpler manifolds with smaller curvature magnitude.
>
> These observations come from systematic experiments, but they remain a conjecture. A natural next step would be to construct manifolds with controlled curvature profiles and investigate extremal cases to refine these bounds.
>
> Again, thank you for the review and let us know if you need more information.

---

### Official Review · Reviewer_UrPE · 2025-10-28

**Soundness:** 2
**Presentation:** 2
**Contribution:** 1
**Rating:** 4
**Confidence:** 4

**Summary:**

The paper investigates how to compute key geometric properties of a data manifold (volume element, scalar curvature, and reach) directly from sampled data points. The authors focus on modeling the manifold as the product of a 1-manifold and a low dimensional ball. They utilize two manifold fitting approaches (MMLS and β-VAE) to empirically test the theoretical bounds proposed by Genovese et al. and Fefferman et al.

**Strengths:**

The paper addresses a critical challenge in learning theory: the difficulty of empirically validating theoretical bounds based on manifold assumptions due to the unknown properties of real-world data manifolds. As a first attempt to bridge this gap, the authors propose a straightforward experimental framework designed to test such theoretical bounds. This initiative provides a valuable starting point for experimentally probing the applicability and limitations of manifold-based theoretical results.

**Weaknesses:**

1. Limited novelty. While this appears to be the first framework for empirically testing manifold bounds, it primarily integrates well-established techniques, ranging from manifold fitting to computing geometric metrics. The potential for reproducibility and generalization to different types of manifolds is limited, which further restricts the novelty of the approach.

2. Low generalizability: The proposed method is confined to manifolds that are the product of a 1-manifold and a low-dimensional ball. This is a strong assumption that does not extend well to the diversity encountered in real-world datasets.

3. Relevant prior work not discussed. he paper omits discussion of several important lines of related research, such as bounds on manifold reconstruction [1,2,3] and bounds built on similar manifold assumptions[4].

[1] Partha Niyogi, Stephen Smale, and Shmuel Weinberger. Finding the homology of submanifolds with high confidence from random samples. Discrete & Computational Geometry, 2008.

[2] Hariharan Narayanan and Sanjoy Mitter. Sample complexity of testing the manifold hypothesis. In Advances in Neural Information Processing Systems, 2010.

[3] Stefan C. Schonsheck, Jie Chen, and Rongjie Lai. Chartauto-encoders for manifoldstructured data. CoRR, abs/1912.10094, 2019.

[4] Yao, J., Goswami, M., & Chen, C. (2024). A theoretical study of neural network expressive power via manifold topology. arXiv preprint arXiv:2410.16542.

**Questions:**

Clarification needed regarding bound evaluation. The paper does not explicitly present the two evaluated bounds (expand the computation of constants), neither in the main text nor the appendix. since the role and value of the constant C are not well explained, it is difficult to interpret the computational details. It is unclear how the authors compute Genovese's bound and plot it alongside Fefferman's bound. Are these bounds presented on the same scale for direct comparison?

---

> ### Author Response · Authors · 2025-11-26
> **Response to review**
>
> We thank you for the review. Below are some answers to your questions.
>
> **Limited novelty**
>
> Our contribution is not a new manifold-learning algorithm but a practical framework for empirically evaluating geometric assumptions underlying theoretical bounds. There is currently a clear gap between (i) analytically defined manifolds with known geometry but limited realism, and (ii) real datasets where geometric quantities cannot be measured with verifiable precision. Our datasets occupy the middle ground: simple enough that geometric quantities can be estimated accurately via finite differences, yet rich enough to model a large variety of controlled scenarios.
>
> Although the generative families are low-dimensional products, many structurally different cases can be explored by varying the underlying image generator (e.g., COIL-20 subsets, rotating-chair datasets, multi-object scenes, occlusions, different sprite geometries, etc.). The novelty lies in producing reliable ground-truth-like geometric evaluations for a broad range of controlled settings, which has not been available before and is essential for stress-testing manifold-based bounds.
>
> **Low generalizability**
>
> The restricted intrinsic dimension is a design choice, not a limitation of the method’s intent. Our goal is to provide benchmark datasets with accurate geometric estimates, not a general-purpose geometric estimator for arbitrary manifolds. The framework is meant to serve as a diagnostic and benchmarking tool for theory and models, not as a replacement for general manifold-learning techniques.
>
> **Relevant prior work**
>
> We expanded the related-work section to include the lines of work you mentioned, in addition to other publications that recently  fell to our attention.
>
> **Clarification of bound evaluation**
>
> We substantially revised the section on evaluating the Genovese et al. and Fefferman et al. bounds. The constants in the bounds are not directly computable, so we fit exponential curves to the empirical error–scale relationship and place upper/lower bounds just above/below the fitted curve, following the standard practice in empirical minimax-rate estimation. Appendix C.2 contains now a lot of information on that.
>
> Thank you again, please let us know if we missed something.

---

### Official Review · Reviewer_pro4 · 2025-10-30

**Soundness:** 2
**Presentation:** 2
**Contribution:** 2
**Rating:** 2
**Confidence:** 3

**Summary:**

The paper brings forward a new mechanism to evaluate theoretical bounds in a rich empirical setup. They first present two previous theoretical results on the sample complexity of learning data distributions with support on low-dimensional manifolds embedded in higher-dimensional Euclidean spaces. They introduce a set of finite-difference methods to estimate three measures: curvature, reach and volume. They finally present quantitative results of their verification pipeline which lets them comment on which of the two theoretical bounds are tighter.

**Strengths:**

The paper presents a novel idea and does a good job developing a framework to verify theoretical results using empirical methods on complex manifolds. Their empirical validation of the two bounds is impressive and novel, as far as I can tell.

**Weaknesses:**

One of the main weaknesses of the paper is that it is missing some key details. I am unclear as to how MMLS works or how $\beta$-VAE fit syntehtic data from the content of the paper. How does the sampling technique described in lines 266-272 ensure that you "obtain more uniform subsets"? What is dist in "To compare with theoretical rates, we regress log(dist) against log(n)"? (Line 359). This is a key detail because it tells us how your comparison to theoretical rates is valid. Please point me to the regression loss curve. Your second application of Manifold analysis (line 414) is not listed as a key result and I am not certain how this contributes to your goal of bridging the gap between theory and practice. I am also not able to grasp your conclusion on why $\beta$-VAE struggles to converge: "latent manifold emphasizes semantic clustering at the cost of geometric distortion". These are some issues that could be addressed to help me understand the manuscript better. Could you also comment on the hyper-parameters you searched for $\beta$-VAE?

Could you show independence of the Hausdorff Distance wrt the ambient dimension as claimed by the two bounds you have presented? This discussion was lost even though this is mentioned in your review of the theoretical results. I would also be curious in how the differences in $r,s$ manifest in the Hausdorff distance.

Another issue I found was that comparison to existing methods is a bit lacking. The finite-difference estimators are a major contribution of yours, I believe that the reach estimate follows from Aamari et al 2019 and similarly there has been past work on curvature estimates in the field of complex networks [1]. I am not discouraging the authors from using their suite to make these measurements but there ought to be a comparison to existing techniques.

I encourage the authors to rectify these issues because I find their approach of connecting theory to practice incredibly promising. I believe some key details are missing and the broader community would benefit from the authors presenting them in the main body of the paper.

----------------------------------------

References:

[1] Comparative analysis of two discretizations of Ricci curvature for complex networks, Areejit Samal, R. P. Sreejith, Jiao Gu, Shiping Liu, Emil Saucan & Jürgen Jost

**Questions:**

I have the following minor issues and questions:

Line 130: In definition of G what are $n_1, ..., n_d$? I would introduce these here if these are some form of scaling parameters.

Line 178: You might be overloading $d$ here, using it both for distance and dimensionality. Furthermore, how is this distance defined?

Appendix A.1 and A.2 seem to be duplicates of the geometry background section (Section 3) in the main paper. I would expect the authors to de-duplicate it for a concise presentation of their work.

Line 180: what is "scalar curvature integral"? I dont see this defined.

Line 163-167: I believe these lines are using the Einstein notation, I think the authors should at least mention this in their work. If it has been specified and I missed it please point me to the section.

Thoerem 1: where is the idea of fiber introduced in the paper? I understand that the some of the intended audience might know this but I believe it might be helpful for the broader audience for the authors to define it in their work or explain it informally.

Theorem 1: What is $\mathcal Q$ here? Are Genovese et al reasoning about the minimax risk over a family of distribution?

Line 347: sub-section 3 -> section 3

Line 304-305: "All three estimators have been tested on families of manifolds with known closed-form quantities" could you please point to these plots and description of the experiments?

Figures 1 and 2 seem to be very similar, I am not sure what is the value addition from adding this schematic in Figure 2.

---

> ### Author Response · Authors · 2025-11-26
> **Response to review**
>
> We would like to thank you for the detailed review. Below we try to address your concerns.
>
> **Missing key details**
>
> This is a fair concern. The framework contains many components, and including all details in the main paper is not feasible within space constraints. We expanded the supplementary material to fill the gaps you identified. Key additions:
>
> *MMLS and $\beta$-VAE details*
>
> We added a clearer explanation of MMLS, including its role in geometric approximation and the way we use it, we also specify the $\beta$-VAE architecture, training hyper-parameters, and its purpose in fitting synthetic data. Those can be now found in Appendix C.1.
>
> *More uniform subsets*
>
> Direct sampling is computationally expensive, so we instead sample from the dense dataset grid with probabilities proportional to the estimated volume element. This yields a discrete approximation to uniform volume sampling. Again, Appendix C.1 contains now this information.
>
> *Definition of “dist” and regression details*
>
>  We reorganized this section and added a precise definition of the distance quantity used in the log–log regression. The supplementary now includes the regression curves, residuals, and a description of the fitting procedure. Those are in Appendix C.2.
>
> *Manifold analysis (line 414)*
>
> This corresponds directly to our third contribution (“tracing how learned representations reshape data geometry”). It illustrates how geometric structure evolves through a network, which is crucial for connecting empirical behavior with geometric assumptions in theoretical bounds.
>
> *Interpretation of the $\beta$-VAE results*
>
> The $\beta$-VAE converges but generates blurry reconstructions. Tracking curvature and inter-class distances across layers reveals that curvature magnitude increases while class manifolds separate more strongly near the latent bottleneck. This could suggest:
>
> The KL regularization compresses manifolds toward a low-dimensional Gaussian, reducing local geometric fidelity (leading to blur resembling small pose/rotation changes).
>
> At the same time, the representation increasingly emphasizes semantic separation between classes over preservation of fine-grained geometric transformations.
>
> This motivates the inclusion of intermediate-layer geometric analysis: it highlights where and how geometric distortion accumulates inside the encoder and decoder.
>
> *Hyper-parameters for $\beta$-VAE*
>
> We also added the hyper-parameters and training details in Appendix C.1. We use almost the same values used by a repository which implements $\beta$-VAE. Then the same values are used on all experiments.
>
>
> **Ambient-dimension dependence of the Hausdorff distance**
>
> The Genovese and Fefferman bounds state that the minimax rate depends on intrinsic geometry, with ambient dimension affecting constants but not the scaling exponent. A full experimental ablation would require restructuring the datasets, and due to scope and length constraints we did not include it. We acknowledge this as a good suggestion for future work.
>
> **Differences in r and s**
>
> Likewise would require additional ablations we would like to leave this for future experiments.
>
> **Comparison to existing geometric estimators**
>
> We extended the discussion on reach and curvature estimators in the related work (section 2). There we added the work you mention and some more bibliography. Most notably a paper we recently discovered, "Optimal reach estimation and metric learning", which is newer work from Aamari et al. with a better estimation of the reach.
>
> Below are the responses to the questions.
>
> **Line 130**:
>
> The parameters correspond to the grid resolution along each generative dimension, i.e. $n_1\cdot\ldots \cdot n_d = n$
> We now define this explicitly in that section.
>
> **Line 178**:
>
> We modified notation to avoid overloading and also added clarifications. This distance is the Euclidean distance of a point $p$ to a set $A$.
>
> **Appendix A.1/A.2 duplication**:
>
> We removed duplicate explanations and tightened the geometry review to avoid redundancy.
>
> **Line 180 (“scalar curvature integral”)**
>
> We now define this formally in a footnote as the integral of the scalar curvature over the manifold, consistent with differential-geometric conventions.
>
> **Lines 163–167 (Einstein notation)**
>
> We added a brief note indicating the use of Einstein summation.
>
> **Theorem 1 (“fiber”)**
>
> We added a definition of it.
>
> **Theorem 1 ($\mathcal{Q}$)**
>
> Indeed, this is a family of distributions defined around the manifold. We added a definition.
>
> **Line 347**
>
> Fixed.
>
> **Line 304-305, tests**
>
> We have included now our codebase to the submission. The tests are under `microscope/computations_grid/tests` while the definitions of the manifolds and the code for sumbolically computing their properties is under `microscope/manifold_examples`
>
> **Figure 1 vs. Figure 2**
>
> You have a good point. We removed figure 2 to save space.
>
> Thank you again for the review, it help a lot. I hope we answered most of your questions.

---

### Official Review · Reviewer_wh6q · 2025-11-01

**Soundness:** 2
**Presentation:** 2
**Contribution:** 2
**Rating:** 4
**Confidence:** 3

**Summary:**

This paper presents a systematic empirical–geometric framework for studying the data manifold hypothesis under controlled settings. The authors build dense-grid versions of common benchmark datasets (extended dSprites, extended COIL-20) that allow near-continuous sampling of transformation parameters, enabling finite-difference (FD) estimation of geometric quantities such as volume, curvature, and reach. These ground-truth-like quantities are then used to (i) benchmark classical manifold-fitting bounds (Fefferman, Genovese, etc.) and (ii) analyze layer-wise geometry evolution in β-VAEs.

Overall, the paper is well executed and the results are clear and interpretable. The main contribution lies in the framework—combining dense-grid datasets with FD-based geometric estimation and empirical validation of theoretical bounds. While each element builds on existing ideas, the integration into a reproducible benchmark pipeline is valuable.

This is a well-written and methodologically sound paper whose main contribution is an integrated, reproducible pipeline for empirically probing data-manifold geometry and testing theoretical bounds. While the individual ingredients are not new, the overall framework fills a gap between theory and empirical practice. Clarifying the treatment of theoretical constants, explicitly positioning the work against prior reach/curvature studies, and open-sourcing the pipeline would strengthen both its novelty and community value.

**Strengths:**

* Solid motivation and clear presentation: The paper provides a coherent workflow from dataset construction to theoretical benchmarking. Figures are easy to interpret, and the connection between geometric theory and empirical practice is well articulated.
* Technically sound methodology: The finite-difference approach and reach estimation procedures are standard but implemented carefully, with convincing O(h^2) convergence plots.
* Bridging theory and experiment: Using real image manifolds to test reach/volume-based bounds is a refreshing way to connect classical geometry with representation learning.
* Potential for community impact: If released as a package, the proposed framework could serve as a valuable benchmark for manifold estimation and representation-geometry research.

**Weaknesses:**

* Limited novelty beyond integration: The key contribution appears to be the framework rather than a fundamentally new theoretical or algorithmic idea. The paper would benefit from a more explicit discussion of how it extends beyond existing works on curvature or reach estimation (e.g., Aamari et al. 2019), theoretical bounds (Fefferman et al.), and prior controlled datasets (dSprites, COIL-20).
* Choice of constants in theoretical comparisons: In Figures 5–7, empirical curves are plotted alongside theoretical upper/lower bounds. However, it is unclear how the constants in those bounds were chosen or fitted. Were they analytically derived, or manually scaled for visualization? Explicitly stating this would clarify the strength of the comparison.
* Open-source and reproducibility: Since the main value lies in the pipeline, the paper would be significantly strengthened by releasing the dataset generators, FD estimators, and plotting utilities as an open-source package.
* Interpretation of mixed bound-matching behavior: In some regimes, the empirical curves align with the theoretical upper bound, while in others they hug the lower bound. It would be interesting to discuss why this happens—e.g., differences in curvature concentration, sampling density, or estimator bias.
* Scalability and scope: The FD approach depends on dense grids, which may not scale beyond low-dimensional transformations. A short discussion of this limitation would help readers understand the generality of the framework.

**Questions:**

1. How were the constants in the theoretical bounds chosen when overlaying them on the empirical results?
2. Do you plan to release the dataset-generation and geometry-estimation code as an open-source package?
3. Can you provide intuition for why the empirical scaling sometimes matches the upper bound and sometimes the lower bound?
4. How practical is the dense-grid FD approach for manifolds beyond two or three intrinsic dimensions?
5. How does your framework compare to prior curvature/reach-estimation and manifold-fitting pipelines in terms of computational efficiency and required sample density?

---

> ### Author Response · Authors · 2025-11-26
> **Response to review**
>
> We want to thank the reviewer for the comments and the useful suggestions. Below are our responses to the individual points:
>
> **Limited novelty beyond integration**
>
> We agree that our contribution is primarily the integration of known components into a coherent benchmark. This is intentional: prior work on reach or curvature estimation (e.g., Aamari et al. 2019) and theoretical bounds (Fefferman et al.) are designed for general datasets with unknown geometry. In contrast, our setting uses datasets where the geometry is exactly known from the underlying generative parameters, and where finite-difference estimates are accurate enough to act as practical ground truth. dSprites, COIL-20, and related datasets were introduced for studying representation learning, not for extracting precise geometric quantities; our contribution is to repurpose and extend them for this purpose and to build a full, reproducible pipeline around them.
>
> This distinction matters because theoretical bounds often contain constants that are extremely difficult to compute in practice, with hidden dependencies on the manifold’s geometry and ambient dimension. This makes empirical validation nearly impossible in real-world datasets. Our benchmark fills precisely this gap: it provides controlled, realistic examples on which existing methods and bounds can finally be stress-tested and better understood. We have clarified this positioning in the revision.
>
> **Choice of constants in theoretical comparisons**
>
> The constants in the upper and lower bounds are hard to compute analytically and one needs to use approximations which make them less precise. For this reason, we estimate them empirically. We fit an exponential curve to the empirical error as a function of sample size and place the theoretical upper/lower curves just above or below this fitted curve. The supplementary now includes these fits and their residuals (Appendix C.2). We also have an additional section with an analytical computation of the simplest bound, in order to demonstrate the difficulties.
>
> **Open-source and reproducibility**
>
> Yes. Our goal is to release the full framework. The current codebase is now included in the submission, and we plan to open-source it after the decision.
>
> **Mixed bound-matching behavior**
>
> This is a good point. Actually, there seems to be a connection between the integral of the absolute value of the curvature and this behavior, the higher it is, the more do the curves move towards the upper bounds. We added a brief discussion of this intuition in section 5 under "Results". Nevertheless, this is just an observation, one would need a new series of experiments to verify this.
>
> **Scalability and scope**
>
> The FD approach relies on dense grids and is therefore limited to manifolds of low intrinsic dimension (typically up to 4-5). We make this explicit in the paper, especially in the discussion section, emphasizing that the framework is designed as a controlled benchmark rather than a general-purpose estimator.
>
> **Comparison to prior pipelines**
>
> We added a comparison of the complexity of our method with other estimators of the reach and curvature at the end of section 4. We have not performed any experimental comparisons though (e.g. on the toy manifolds with known geometry), as this would involve implementing the corresponding methods.
>
>
> We hope these clarifications address some of your concerns. If there is something we missed, please let us know.

---

### Author Response · Authors · 2025-11-26
**Note to all reviewers**

Thank you all for your input, it helped substantially improved the paper. We revised the main text, expanded the supplementary material, and included the full code for the framework and experiments. Some key updates:

- Our goal is to open-source the framework. While some code improvements remain, we plan to release it upon the paper decision. For now, the complete code is included in the submission for inspection and reproducibility. Should you have any issue accessing it, please let us know. *Please note one can download the code from the tiny `zip` icon next to `Supplementary Material`*.
- We added a detailed explanation of how we estimate the constants in the bounds. These are not computed directly; instead, we fit an exponential curve to the data and place upper and lower bounds just above or below this fit. The supplementary now includes an evaluation of the fit quality.
- We clarified that our main contribution is to provide benchmarking and exploration tools, which are intentionally designed for datasets with small intrinsic dimension.
- The supplementary now contains additional 3D projections of the datasets and pointwise visualizations of the geometric quantities, to offer more intuition.

For details addressing specific questions and concerns, please refer to the individual responses. We encourage you to have a second look at the updated paper and the code made available.

Thank you again for your time.

---

### Author Response · Authors · 2025-12-01
**Message to the Area Chair**

Dear Area Chair,

Due to the leak incident, we understand that the reviewers’ post-leak feedback cannot be considered and that the final decision will be based on input submitted before that point. We would like to respectfully highlight one detail that may help contextualize the reviewers’ original scores.

A substantial portion of the reviewers’ concerns centered on missing information, incomplete explanations, and the absence of code. Before the leak occurred, we addressed these issues comprehensively:

- The full code for the framework and experiments is now included in the submission for inspection and reproducibility.
- The main text and supplementary material have been significantly expanded, including detailed explanations of the bound-constant estimation procedure and additional geometric visualizations.
- Several ambiguities raised in the original reviews have been clarified, including definitions and methodological steps.

Given the circumstances, we would be grateful if you could consider these revisions and the now-available codebase when forming your independent assessment.

Thank you for your time and for handling this situation under challenging conditions.

---

### Meta-Review · Area_Chair_caFf · 2026-01-06

**Summary:**

This paper proposes a benchmark to empirically test learning theory bounds derived under the assumption of the manifold hypothesis. Reviewers raised several core concerns, including a lack of novelty, the type of considered manifolds being too narrow, and a lack of scaling. Although the authors addressed several minor clarity concerns in their rebuttal, I do not believe that the more fundamental concerns the reviewers raised were addressed in the rebuttal, and I thus recommend rejection.

**Reviewer Concerns:**

I believe some minor phrasing concerns regarding the main goals and contributions of the paper were addressed. The lack of accompanying code was also addressed, and clarifications about figures were also addressed. As I said in the summary though, I do not believe the core concerns of lack of novelty, the type of considered manifolds being too narrow, and a lack of scaling were addressed though.

**Reviewer Scores:**

I do not believe reviewers would have increased their scores.

---

### Decision · Program_Chairs · 2026-01-26

Reject